# Prediction of Non-Response to Neoadjuvant Chemoradiotherapy in Esophageal Cancer Patients with ^18^F-FDG PET Radiomics Based Machine Learning Classification

**DOI:** 10.3390/diagnostics12051070

**Published:** 2022-04-24

**Authors:** Roelof J. Beukinga, Floris B. Poelmann, Gursah Kats-Ugurlu, Alain R. Viddeleer, Ronald Boellaard, Robbert J. de Haas, John Th. M. Plukker, Jan Binne Hulshoff

**Affiliations:** 1Department of Nuclear Medicine and Molecular Imaging, University Medical Center Groningen, University of Groningen, Hanzeplein 1, 9700 RB Groningen, The Netherlands; j.beukinga@gmail.com (R.J.B.); r.boellaard@umcg.nl (R.B.); 2Department of Surgical Oncology, University Medical Center Groningen, University of Groningen, Hanzeplein 1, 9700 RB Groningen, The Netherlands; florispoelmann@gmail.com (F.B.P.); j.t.m.plukker@umcg.nl (J.T.M.P.); 3Department of Pathology, University Medical Center Groningen, University of Groningen, Hanzeplein 1, 9700 RB Groningen, The Netherlands; g.kats-ugurlu@umcg.nl; 4Department of Radiology, University Medical Center Groningen, University of Groningen, Hanzeplein 1, 9700 RB Groningen, The Netherlands; a.r.viddeleer@umcg.nl (A.R.V.); r.j.de.haas@umcg.nl (R.J.d.H.); 5Department of Radiology and Nuclear Medicine, Cancer Center Amsterdam, Amsterdam UMC, Vrije Universiteit Amsterdam, De Boelelaan 1117, 1081 HV Amsterdam, The Netherlands

**Keywords:** esophageal neoplasms, neoadjuvant therapy, positron-emission tomography

## Abstract

Background: Approximately 26% of esophageal cancer (EC) patients do not respond to neoadjuvant chemoradiotherapy (nCRT), emphasizing the need for pre-treatment selection. The aim of this study was to predict non-response using a radiomic model on baseline ^18^F-FDG PET. Methods: Retrospectively, 143 ^18^F-FDG PET radiomic features were extracted from 199 EC patients (T1N1-3M0/T2–4aN0-3M0) treated between 2009 and 2019. Non-response (*n* = 57; 29%) was defined as Mandard Tumor Regression Grade 4–5 (*n* = 44; 22%) or interval progression (*n* = 13; 7%). Randomly, 139 patients (70%) were allocated to explore all combinations of 24 feature selection strategies and 6 classification methods towards the cross-validated average precision (AP). The predictive value of the best-performing model, i.e AP and area under the ROC curve analysis (AUC), was evaluated on an independent test subset of 60 patients (30%). Results: The best performing model had an AP (mean ± SD) of 0.47 ± 0.06 on the training subset, achieved by a support vector machine classifier trained on five principal components of relevant clinical and radiomic features. The model was externally validated with an AP of 0.66 and an AUC of 0.67. Conclusion: In the present study, the best-performing model on pre-treatment ^18^F-FDG PET radiomics and clinical features had a small clinical benefit to identify non-responders to nCRT in EC.

## 1. Introduction

Most patients with locally advanced esophageal cancer (T1N1-3M0/T2–4aN0-3M0) benefit from neoadjuvant chemoradiotherapy (nCRT) followed by esophagectomy [1]. After nCRT, 29% of these patients have a pathologically complete response and 32% have a near-complete response with < 10% vital tumor cells [1]. However, a substantial group of patients does not respond: 8% develop progressive disease usually as interval metastases, and 18% only achieve a limited response (i.e., >50% remaining vital tumor cells) [1,2]. Patients with poor response to nCRT followed by complete resection have a similar prognosis as those who undergo primary esophagectomy [3]. Pre-treatment identification of non-responders would allow for alternative treatment strategies, e.g., earlier surgical intervention or additional targeted therapies. This custom-based approach may prevent prolonged useless exposure to nCRT with potential risk of radiation-induced toxicity or tumor expansion with delay of surgery.

There is increasing evidence that intratumoral heterogeneity is a major determinant of non-response to nCRT. Heterogeneity on the cellular level can be caused by genetically distinct subpopulations for sustained tumor growth, including cancer stemness, genetic diversity in ligand/receptor expression, tumor microenvironment with metabolic reprogramming, and epigenetic alterations [4,5,6]. As such, it may be caused by distinct subclonal populations with specific patterns of oxygen consumption, glucose metabolism, and cellular proliferation as reflected by subtle spatial variations on medical images [6,7,8]. Although these variations are difficult to detect during regular radiological reading, they may be revealed by voxel-wise pattern recognition techniques such as radiomics [6,7,8,9]. Radiomics phenotyping is a non-invasive analysis which encompasses image acquisition followed by high-throughput extraction of quantitative features from regions-of-interest defined on medical images. These predefined features capture geometric, intensity, and textural information about the tumor and provide a huge amount of non-invasive imaging biomarkers which can be modeled using machine learning classifiers to predict treatment response and prognosis.

Several studies reported promising results in predicting complete response to nCRT when using pre- and/or post-nCRT radiomics derived from CT and/or ^18^F-FDG PET [10,11,12,13,14,15,16,17,18,19]. The innovative field of radiomics might provide similar opportunities in managing non-responding esophageal cancer patients. A great benefit using ^18^F-FDG PET radiomics is the unique ability to provide whole-body quantitative information of spatial phenotypic variation in metabolism, thereby capturing tumor site-related information about tumor resistance to nCRT such as hypoxia, necrosis, and cellular proliferation [20].

However, studies on radiomics are based on complex statistics and analyses, and truly clinically relevant findings are still lacking. Since it is important to transfer knowledge from scientific research more early into real time practice, defining current position using a relative large number of uniform staged EC patients could add to a critical sound view before being suitable in prospective studies. Therefore, the aim of this study was to construct a useful model combining clinical information and radiomic features from pretreatment ^18^F-FDG PET scans to predict non-response to nCRT in esophageal cancer.

## 2. Materials and Methods

### 2.1. Patients

This retrospective study was conducted in accordance with the Dutch guidelines for retrospective studies and rules of the local institutional ethical board, the local ethical board waived the requirement to obtain informed consent (METc 202000093). Between January 2009 and August 2019, 199 patients with locally advanced esophageal cancer (T1N1-3M0/T2–4aN0-3M0) treated with nCRT at our institution were included (Figure 1 displays the inclusion and exclusion criteria). Data collection and reporting of analysis was performed according to the STARD guidelines.

### 2.2. Staging and Treatment

Patients were staged with a thoraco-abdominal CT (Biograph mCT 4–64 PET/CT; Siemens, Erlangen, Germany), ^18^F-FDG PET/CT (Biograph mCT-64 PET/CT; Siemens, Knoxville, TN, USA), and endoscopic ultrasound. Patients were discussed in the multidisciplinary upper gastrointestinal tumor board and treated according to the CROSS regimen (5 cycles of carboplatin (2 mg∙min∙mL^−1^) and paclitaxel (50 mg/m^2^) with 41.4 Gy in 23 fractions) [1]. Restaging was performed 6–8 weeks after nCRT with CT (before 2014) or ^18^F-FDG PET/CT (after 2014). Surgical treatment consisted of a minimally invasive or open transthoracic esophagectomy.

### 2.3. Histopathologic Response Evaluation

Two experienced pathologists assessed the resected surgical specimen and scored response on the five-point Mandard tumor regression grade (TRG) scale. Patients were considered non-responders if residual tumor was scored as TRG 4 (i.e., fibrosis and tumor cells with preponderance of tumor cells), as TRG 5 (i.e., tumor tissue without signs of regression), or if progressive disease was detected at restaging or during surgery [21].

### 2.4. PET/CT Imaging

After at least six hours of fasting, all patients received 3 MBq/kg ^18^F-FDG 60 min prior to imaging. Low-dose CT (80–120 kV; 20–35 mAs; and 5 mm section thickness) and PET images (voxel size 3.1819 mm × 3.1819 mm × 2 mm and 2–3 min scans per bed position) were acquired in radiation treatment planning position. To harmonize SUV (standardized uptake value), images were reconstructed in compliance with either NEDPAS or EARL protocols [22].

### 2.5. Tumor Delineation and Radiomic Feature Extraction

Primary tumors were initially delineated on axial images of the baseline PET scans using SUV thresholding with an in-house delineation tool built in MeVisLab (MeVis Medical Solutions AG, Bremen, Germany; version 3.1.1). To optimize the quality of the delineations, the volume of interest (VOI) was manually corrected using both the CT and PET scan in consensus between the collaborating investigators (RJB and JBH). PET scans and corresponding VOI were resampled to isotropic voxel-dimensions of 2 mm × 2 mm × 2 mm using trilinear interpolation. The interpolated VOIs were rounded to binary images. From each VOI, 143 ^18^F-FDG PET-derived radiomic features were extracted with software developed in Matlab 2018b (MathWorks Inc, Natick, MA, USA). Image processing and feature extraction were performed in compliance with guidelines provided by the Image Biomarker Standardization Initiative [23]. To reduce both computational workload and the impact of image noise, textural features were extracted from discretized image stacks (X_discretized = ⌊X_SUV/0.25⌋ + 1). From these discretized image stacks, the spatial distribution of gray-level intensities was scored in three dimensions (26-voxel connectivity) into a single merged texture matrix from which the textural features were calculated.

### 2.6. Radiomics Machine Learning Pipeline

Figure 2 shows the machine learning pipeline written in Python 3 using the open-source machine learning library Scikit Learn (version 0.22.1). Radiomic features entered the machine learning pipeline together with clinical features (histology, clinical T- and N-stage). All continuously scaled features were normalized. If a feature distribution had a skewness between −0.5 and 0.5, the feature was robustly normalized by removing the median and scaling to the interquartile range. In case of an absolute skewness > 0.5, the feature was transformed by a Yeo-Johnson power transformation (which applies monotonic transformations to make the data more Gaussian-like). The data were randomly divided into a training (70% of the samples) and validation test subset (30% of the samples), with preservation of the original response distribution.

To prevent overfitting, i.e., when the model unintentionally is learning noise instead of the underlying trend of the data, the feature space was reduced in four consecutive feature selection steps as illustrated in Figure 2 (resulting in 24 unique feature selection strategies). All combinations of these feature selection strategies were explored with six classification methods (resulting in 144 different machine learning strategies) in terms of model performance on the training and test subsets. In the first feature selection step, we eliminated radiomic features with a low multivendor reproducibility identified by earlier research (i.e., intraclass correlation coefficient ICC < 0.6) and radiomic features with a high Pearson correlation (ρ > 0.8) with conventional features (volume, SUVmax, SUVpeak, SUVmean, and/or total lesion glycolysis) [24]. In the second feature selection step, principal component analysis was used to further reduce dimensionality by creating new uncorrelated variables (i.e., principal components) from the original dataset. The first principal components (sorted by the amount of variance in the data), which explained > 95% of the total variance in the data, were selected for further analysis. Thirdly, to rank features, the impact of 6 univariable filter methods (logistic regression, ANOVA, Fisher score, Relief, T-score, and Gini index) was investigated. The final feature selection step involved incorporation of the least absolute shrinkage and selection operator (LASSO), a regularization technique which simultaneously prevents overfitting and performs feature selection. After feature selection, six different machine learning classifiers were trained: logistic regression, support vector machine, random forest, Gaussian naive Bayes, neural network, and K-nearest neighbors. A 2-fold cross-validation was repeated 5 times in the training subset to tune hyperparameters of the filter methods, regularization, and classifiers, and to select the best-performing model based on the mean average precision (AP) over the different folds. AP measures the area under the precision–recall curve, which describes the trade-off between precision (i.e., positive predictive value) and recall (i.e., sensitivity). The AP metric is particularly useful in this study as it only considers the positive class (minority of the cases) and is unconcerned of the true negatives (majority of the cases). The AP adds statistical value when it exceeds the percentage of non-responding patients in the test subset. To further improve the performance, we used a soft-voting rule classifier which aggregates the predictions of the 10 best-performing models by averaging the class–probabilities of these models. The generalization performances of all models were evaluated on the independent test subset.

## 3. Results

### 3.1. Patients Characteristics

Patient and tumor characteristics of the response and non-response group are summarized in Table 1. Among the included 199 patients, 57 (29%) were non-responders; 39 (68%) had TRG 4 and 5 (9%) TRG 5. Progressive disease was detected at restaging in ten patients (5%) and intraoperatively in three (2%) patients. All these 13 patients were not amenable to further surgery and were considered as having ≥ TRG 4 based on macroscopic progressive tumor. Of all patients, 139 (70%) were allocated to the training and 60 (30%) to the test subset. There were no significant differences in patient characteristics between the training and test subset.

### 3.2. Feature Normalization and Preselection

In total, 143 radiomic and 3 clinical features (clinical T- and N-stage, and tumor histology) entered the machine learning pipeline. In the preselection step, 22 of the 143 extracted radiomic features had a low multivendor reproducibility according to the definitions used in previous research (ICC < 0.6) and were eliminated [24]. Among the remaining 121 radiomic features, 25 had an approximately symmetric distribution (skewness between −0.5 and 0.5) and 96 had a moderate to high skew distribution (absolute skewness > 0.5) and were normalized according to the predetermined normalization approach. After normalization, 65 radiomic features were considered redundant and subsequently removed because of a high Pearson correlation (ρ > 0.8) with one or more conventional features (volume, SUVmax, SUVpeak, SUVmean, and total lesion glycolysis), leaving 3 clinical and 56 radiomic features for further analysis. Appendix A displays the excluded vendor-dependent and -redundant features. The degree of redundancy between the remaining 56 radiomic features is demonstrated by a correlation heatmap with dendrograms (Figure 3). Of these features, 84% had at least one absolute pair-wise Pearson correlation > 0.8, indicating a substantial amount of feature redundancy remaining in the dataset. This redundancy was attempted to be reduced by the subsequent feature selection steps in the machine learning pipeline.

### 3.3. Model Selection and Performance

Figure 4 shows the cross-validated model performance of the 10 best performing models selected from the training subset, based on the AP metric. Model 1–4 were essentially identical models with the same features and identical performances but were constructed through four different machine learning strategies. All these models were support vector machine classifiers trained on the same five principal components, generated by principal component analysis during feature selection. However, these principal components were selected by four different feature selection methods, i.e., relief, ANOVA, logistic regression, and T-score. These models showed an AP (mean ± SD) of 0.47 ± 0.06 on the training subset and were externally validated in the test subset with an AP of 0.66 (Figure 5) and an area under the ROC curve (AUC) of 0.67. This AP is substantially higher than the AP of random classification (percentage non-responding patients in the test subset = 0.28). The learning curve in Figure 6 shows that the training and test AP scores did not fully converge to a point of stability yet, and therefore the model would slightly benefit from more training data. The soft-voting rule classifier, aggregating the predictions from the 10 best-performing models, showed an AP of 0.64 and an AUC of 0.68 on the test subset.

To determine the effect of data heterogeneity, additional sub-analyses were performed on clinical T-stage and histology. Patients with clinical stage T1–T2 tumors (*n* = 38) were excluded because radiomic features from smaller volumes are known to be less reliable [25]. The best performing model in the T3–4a patient group showed a train-AP of 0.47 ± 0.10, a test-AP of 0.48, and a test-AUC of 0.67. Moreover, to increase homogeneity, a separate sub-analysis with only adenocarcinoma patients (*n* = 177) was executed. In this group, the best-performing model exhibited a train-AP of 0.58 ± 0.09, a test-AP of 0.29, and a test-AUC of 0.46.

## 4. Discussion

Following promising results of ^18^F-FDG PET radiomics studies on the prediction of pathologically complete response in esophageal cancer, adequate discriminative ability would be expected in predicting non-response [13,14]. After investigating a wide spectrum of machine learning techniques including data dimension reduction techniques, classifiers, and cross-validated model training, our best performing prediction model was able to learn representative patterns in the dataset (AP 0.66). To test the clinical relevance of this model, only high precisions should be considered within the current clinical scenario as it is extremely important to prevent refrainment of effective nCRT due to false positive predictions. However, the trade-off between recall and precision in this study shows that it is not possible to increase precision without substantially reducing the recall (Figure 6). This would implicate an increase in the number of responding patients that are falsely classified as non-responding patients. Although this study shows a relatively small clinical benefit of combining clinical and radiomic features from pretreatment ^18^F-FDG PET scans, the predictive power is too low to be clinically applicable in predicting non-response to nCRT in esophageal cancer.

We attempted to find the underlying cause of the relatively low predictive ability of this model. The learning curve in Figure 6 indicates that the training process was halted rather prematurely and may slightly benefit from a larger sample size. Furthermore, despite this study was conducted on a relatively homogeneous patient group, two separate analyses were performed to rule out potential influence of remaining data heterogeneity. First, we limited analyses to T3-T4a tumors to reduce the effect of partial volume effects and possible delineation inconsistencies in smaller cancers. Despite this approach being consistent with earlier studies stating that the complementary information of radiomic features substantially increases with larger volumes [26], it did not improve the model performance. Moreover, a separate analysis was performed on adenocarcinomas alone, which respond poorer to nCRT than squamous cell carcinomas [1]. However, this subgroup analysis did not reveal any model improvement either.

So far, several studies investigated temporal changes in ^18^F-FDG PET radiomics features [16,17]. However, as this information can only be extracted after definitive treatment, it has little clinical impact on changing patient management. Tixier et al. did report differences in baseline ^18^F-FDG PET radiomics between non-responders and partial responders, but this study had a small sample size with no external validation [15]. As already known, the difference between training and test performance results emphasizes the necessity of an external validation group and sufficient sample size in order to determine the true predictive value of radiomic models. In addition, differences in imaging features between the groups might be related to only a small but substantial part of distinct subclones, reflecting a crucial area of tumor biology with genomics driven differences.

One of the main issues of radiomics are unestablished measurement errors (i.e., repeatability, reliability, and reproducibility). Moreover, the majority of ^18^F-FDG PET radiomics are sensitive to different sources of variation such as the delineation method, image acquisition, or reconstruction protocols [27,28,29]. In accordance with prior research, a wide range of radiomic features were harmonized by acquiring all scans in a single-center and by using single-vendor settings according to either the “European Association of Nuclear Medicine Research Ltd.” (EARL) compliant reconstruction protocols or “Netherlands protocol for standardization of ^18^F-FDG whole-body PET studies in multi-center trials” (NEDPAS) [22,24]. Due to the retrospective nature of this study, the reconstruction protocol was updated during the course of the study. Only ^18^F-FDG PET radiomic features reliable in a multi-center and multi-vendor setting were preselected for further analysis. Moreover, radiomic features are sensitive to inconsistent tumor delineations due to a great variety in tumor morphology with occasionally blurred tumor margins. Therefore, there is a need for further standardization.

Currently, the prediction of response to nCRT based on only qualitative (traditional subjective reading) and semi-quantitative (e.g., SUV parameters and total lesion glycolysis) baseline and restaging ^18^F-FDG PET data seems to be insufficient. Multiple factors may contribute to an insufficient predictive power due to ^18^F-FDG PET misinterpretation, including proper patient preparation and type of scanner. Besides, staging interpretation may be hindered by esophagitis (e.g., reflux induced, after endoscopic dilatation or radiotherapy), esophageal candidiasis, sarcoidosis, or low-glucose-metabolizing tumors (e.g., mucinous adenocarcinomas) [30]. Beyond ^18^F-FDG PET radiomics, relevant information could be extracted from other functional imaging modalities such as DW-MRI or specific PET tracers. Since PET and MR images capture different intrinsic information about tumor biology, we strongly believe that such a multimodality approach would be able to optimize prediction of non-response to nCRT in esophageal cancer. Imaging information may also be complemented by genomic profiles of esophageal cancers, including data obtained from the Tumor Cancer Genome Atlas. Linking radiomic patterns directly to these genomic profiles, the so-called radiogenomics, may facilitate targeted treatment by radiomics-guided biopsy to specific sites identified with mutational burden or driven mutations.

A next logical step might be the implementation of deep learning algorithms such as convolutional neural networks. Advantages of convolutional neural networks include that features are automatically trained and no predefined handcrafted features are required in order to learn the relationship between input and outcome. Additionally, as tumor delineation is not essential, inconsistencies in delineation methods and intra- and interobserver variability can be reduced, increasing the accuracy. However, to ensure the generalization capability of such studies, even higher sample sizes are required due to the larger number of learnable parameters. This can be a practical limitation and can only be resolved by the standardization of used methods in collected studies, preferably in a multicenter prospective manner and international collaboration [31].

## 5. Conclusions

This is the first study to assess the value of ^18^F-FDG PET radiomics combined with clinical features in the prediction of non-response to nCRT in esophageal cancer. Despite an extensive evaluation using various data dimension reduction techniques, classifiers, and training using cross-validation, we were only able to demonstrate a moderate discriminatory value for the constructed models. In the present study, the clinical impact of the best performing model, containing both clinical and ^18^F-FDG PET-derived radiomic features, was not sufficient to predict non-response to nCRT in esophageal cancer.

## Figures and Tables

**Figure 1 diagnostics-12-01070-f001:**
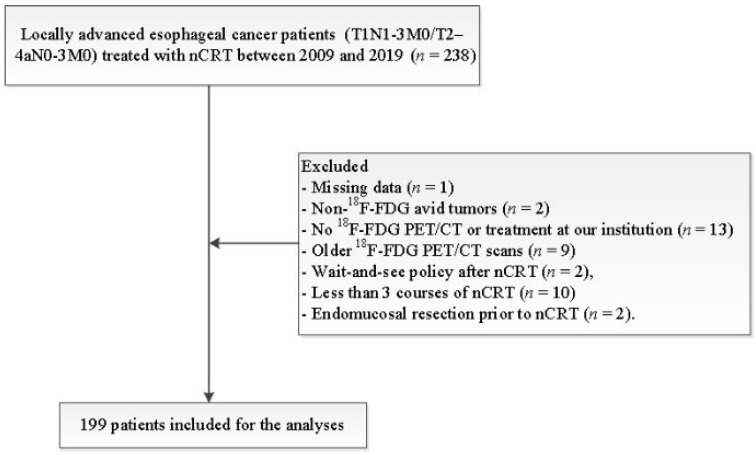
Inclusion and exclusion flowchart. Abbreviations: nCRT = neoadjuvant chemoradiotherapy.

**Figure 2 diagnostics-12-01070-f002:**
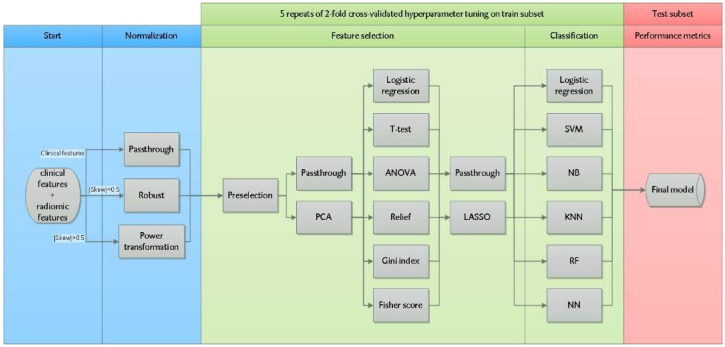
Radiomics machine learning pipeline to train and select a model predicting non-response to nCRT. Radiomic and clinical features were normalized up front (blue area). Hyperparameter tuning was performed on the training subset (green area) with 24 unique feature selection strategies and 6 classification methods. The model with the highest mean average precision (AP) over the different cross validation folds was selected. The performance of this model was tested on the test subset (red area). Abbreviations: Skew = skewness of the distribution, SVM = support vector machine, NB = Gaussian Naive Bayes, KNN = K-nearest neighbors, RF = random forest, and NN = neural network.

**Figure 3 diagnostics-12-01070-f003:**
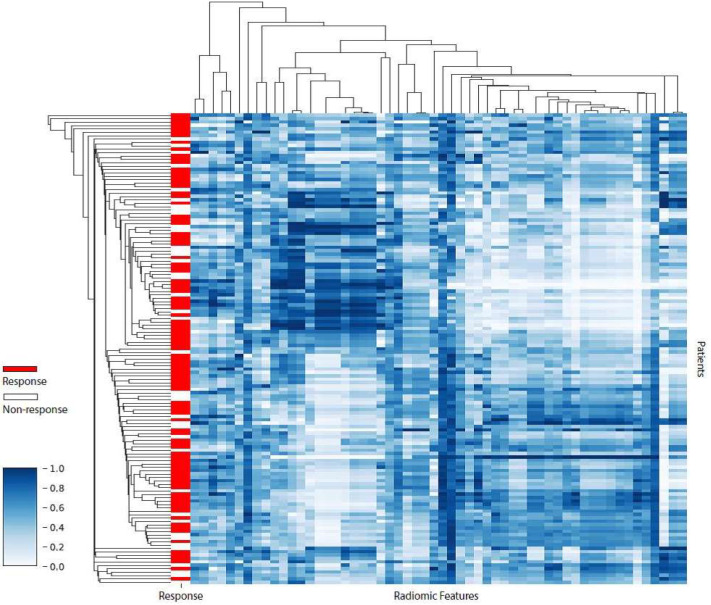
Heatmap revealing radiomic feature clusters with similar expression (standardized on white-blue gradient scale) using unsupervised clustering with Pearson correlation as a measure of similarity. The x-axis represents the preselected radiomic features (*n* = 56) and the y-axis represents esophageal cancer patients in the training subset (*n* = 139). The heatmap reveals a substantial amount of feature redundancy.

**Figure 4 diagnostics-12-01070-f004:**
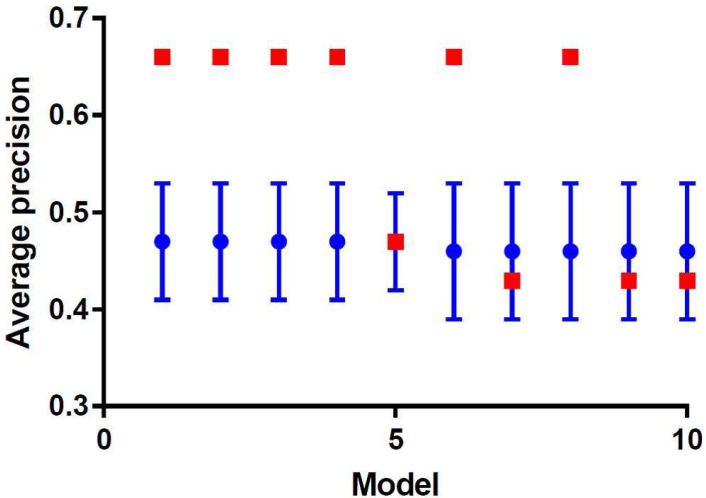
Plot of the 10 best-performing models ordered by the mean average precision over the validation runs in the training subset (blue). The test performance was evaluated on an independent test set (red).

**Figure 5 diagnostics-12-01070-f005:**
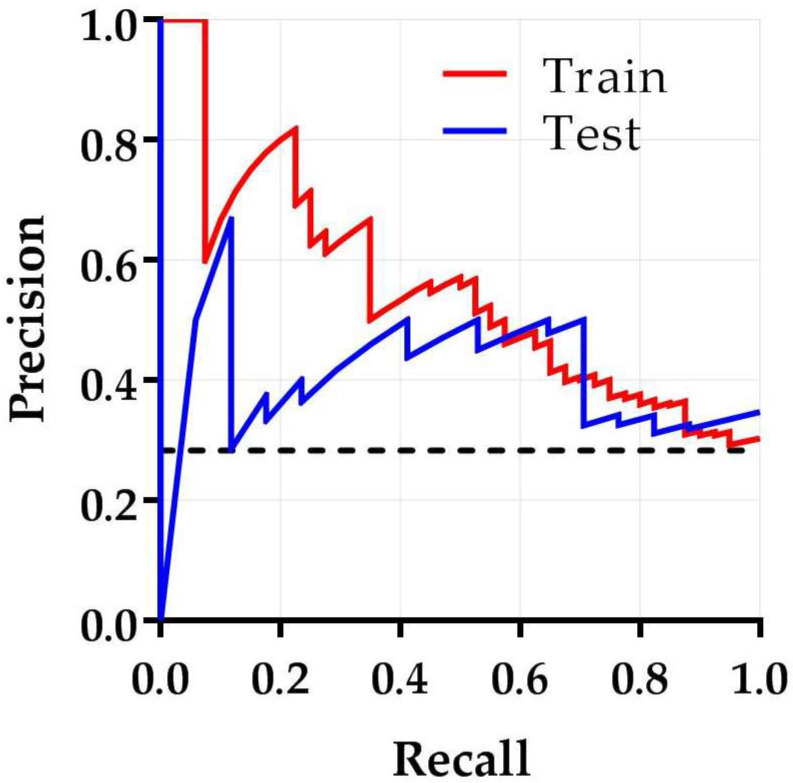
Precision–recall curve of the best performing model demonstrating the trade-off between precision and recall. The area under the precision–recall curve is reflected by the average precision. The average precisions for the training and test subset are 0.47 and 0.66, respectively. The black dashed line is the score of a random classification (0.28).

**Figure 6 diagnostics-12-01070-f006:**
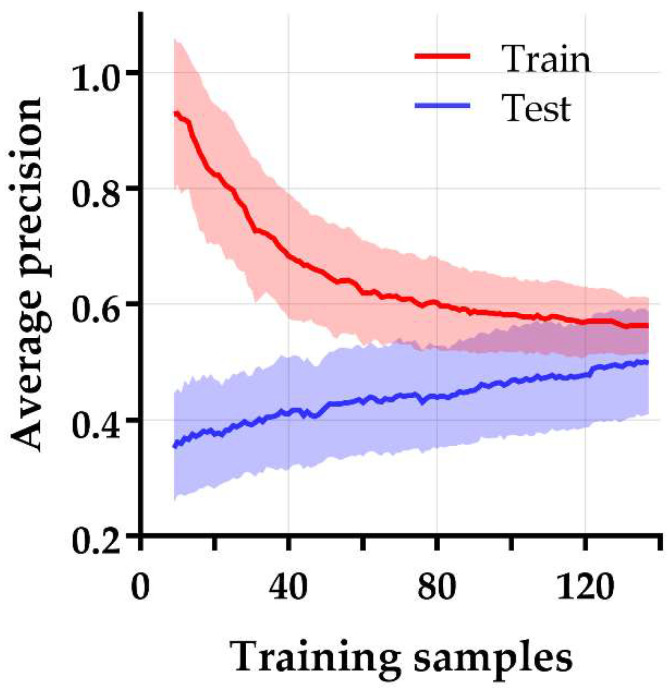
Learning curve of the best-performing model for prediction of non-response after nCRT in esophageal cancer. The average precision is plotted on the y-axis and the number of training samples on the x-axis. The training and test average precision scores did not fully converge to a point of stability yet, suggesting that the training process may slightly benefit from a larger sample size.

**Table 1 diagnostics-12-01070-t001:** Patient and tumor characteristics of responders versus non-responders.

Characteristic	Response (*n* = 142) *n* (%)	Non-Response (*n* = 57) *n* (%)	*p*-Value ^1^
Gender (Male)	113 (79.6)	48 (84.2)	0.446
Age (years), median (IQR)	66 (61–71)	67 (61–72)	0.546 ^2^
Histology Adenocarcinoma Squamous cell carcinoma	124 (87.3) 18 (12.7)	53 (93.0) 4 (7.0)	0.231
Tumor location Mid Distal Gastroesophageal junction	20 (14.1) 96 (67.6) 26 (18.3)	2 (3.5) 42 (73.7) 13 (22.8)	0.057
Tumor length (cm), median (IQR)	6.0 (4.0–7.0)	5.0 (4.0–8.0)	0.595 ^2^
Clinical T-stage T1 T2 T3 T4a	2 (1.4) 28 (19.7) 107 (75.4) 5 (3.5)	0 (0.0) 8 (14.0) 44 (77.2) 5 (8.8)	0.246
Clinical N-stage N0 N1 N2 N3	30 (21.1) 75 (52.8) 33 (23.2) 4 (2.8)	16 (28.1) 23 (40.4) 15 (26.3) 3 (5.3)	0.399
CRM (0 mm) R1 NA ^3^	5 (3.5) 0 (0.0)	3 (5.3) 13 (22.8)	0.371

*Abbreviations*: IQR = interquartile range, CRM = circumferential resection margin, R0 = microscopically tumor-free resection, R1 = microscopically irradical resection, and NA = not applicable. ^1^ Likelihood ratio test. ^2^ Mann–Whitney U test. ^3^ No resection was performed due to distant metastases found before or during surgery.

## Data Availability

Not applicable.

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
