# Peer review of "Prediction of Non-Response to Neoadjuvant Chemoradiotherapy in Esophageal Cancer Patients with 18F-FDG PET Radiomics Based Machine Learning Classification"

_diagnostics, 2022, doi:10.3390/diagnostics12051070_

Round 1
Reviewer 1 Report
This study developed a machine learning pipeline for predicting non-response to neoadjuvant chemotherapy from PET scans in esophageal cancer. The scans, which specifically used the radiotracer, 18F-FDG, were processed through feature selection step to extract the radiomic representations used for prediction. The paper is very well written, the methods are detailed and include adequate description of the data, treatment protocols and evaluation of treatment response. Although the model performance was moderate, which compromised the clinical impact of the paper, the authors have identified several underlying limitations and possible solutions that need to be considered in future work to enable accurate radiomic-based prediction of non-response in esophageal cancer. Therefore, the manuscript may be considered for publication as an exploration of predictive modeling techniques in esophageal cancer treated with neoadjuvant chemotherapy, facilitating multiple future studies in the field. I do recommend the following considerations before publication:
- Paragraph 2 of the Introduction would benefit from some literature about the significance of intratumoral heterogeneity and role of medical images in determining non-response.
- It is notable that the testing AP is better than train AP. Did the authors investigate whether this happens for different data splits?
- Define the acronym SUV at first use
- In terms of model development, please study whether the following could have compromised the model predictive power:
- The feature reduction could be too stringent
- The 2-fold cross validation could be too small to tune the hyperparameters
- The last sentence of the conclusion may give the impression to the reader that 18F-FDG PED radiomic features are inadequate for clinical value, which is not proven yet since CNN for instance could yield a predictive power. Please reform the sentence to make it central about the model rather than the features.
Author Response
Thank you for your review of our article ‘Prediction of Non-Response to Neoadjuvant Chemoradiotherapy in Esophageal Cancer Patients with 18F-FDG PET Radiomics based Machine Learning Classification’, and thank you for your positive comments and thorough questions.
- Paragraph 2 of the Introduction would benefit from some literature about the significance of intratumoral heterogeneity and role of medical images in determining non-response.
We agree with the reviewer that this could be of additional value and therefore we added more information about the significance of intratumoral heterogeneity and role of medical images. - It is notable that the testing AP is better than train AP. Did the authors investigate whether this happens for different data splits?
The authors acknowledge that higher test AP compared to train AP is a somewhat unusual behavior and is most likely caused by a small test sample size. Indeed, different data splits by an outer cross validation probably provide better estimation of the true generalized predictive power. However, in this case there may be several best models and hyperparameters corresponding to each fold. Therefore, we can only measure the predictive power of the machine pipeline and not that of a single model. We therefore chose to disregard the outer cross validation. - Define the acronym SUV at first use
We added the full description of the acronym SUV the first time it is used (line 106-107, page 3). - In terms of model development, please study whether the following could have compromised the model predictive power:
The feature reduction could be too stringent
Feature reduction was part of tuning the machine learning pipeline. Hence, the optimal set of features was selected based on repeated cross validation. There were no constraints on the number of selected features. Furthermore, the features in the final model were independently selected by 4 different filter methods, indicating stability of the machine learning pipeline. It is therefore unlikely that feature reduction was too stringent.
The 2-fold cross validation could be too small to tune the hyperparameters.
There is a trade-off between computational cost required and selecting the optimal model and its hyperparameters. We chose to repeat the 2-fold cross validation 5 times per hyperparameter. This should be sufficient to promote robust hyperparameter tuning (i.e. at least identify which hyperparameters are superior among others), and yielded acceptable computational cost at the same time. - The last sentence of the conclusion may give the impression to the reader that 18F-FDG PET radiomic features are inadequate for clinical value, which is not proven yet since CNN for instance could yield a predictive power. Please reform the sentence to make it central about the model rather than the features.
We agree with the reviewer that our research applies to the best performing model in the present study, and we therefore changed the sentence to be applicable for the best model in the present study (lines 352-355, page 11). Hopefully, other researchers will further elaborate our findings and will also include the CNN for a better predictive model.
Reviewer 2 Report
Dear authors,
your manuscript is well written down, linear and clear. The methods are well depicted and the discussion is sufficiently covered. The conduction of this radiomics study is commendable. I appreciated that you pointed out the limitations of the study and elucidated the relative little clinical impact of the process. I don't have any significant remarks or suggestion to made.
Author Response
We would like to thank the reviewer for his/her positive comments.